# Rapid diagnostic tests, laboratory-based immunoassay and nucleic acid testing strategies for long-acting injectable pre-exposure prophylaxis: A systematic review and meta-analysis

Warittha Tieosapjaroen [1,2,3‡], Eloise Williams [4,5,6‡], Cheryl C. Johnson [7], Carlota Baptista Da Silva [8], Magdalena Barr-DiChiara [9], Michelle Rodolph [7], Heather Leigh Ingold [7], Heather-Marie A. Schmidt [7,10], Mateo Prochazka [7], Busi Msimanga [7], Celine Lastrucci [7], Hortensia Peralta [7,11], Lastone Chitembo [7,12], Precious Andifasi [7,12], Nandi Siegfried [8], Raphael J. Landovitz [9], Jason J. Ong [1,2,13]*

1 Melbourne Sexual Health Centre, Bayside Health, Melbourne, Australia, 2 School of Translational Medicine, Monash University, Melbourne, Australia, 3 Kirby Institute, University of New South Wales, Sydney, Australia, 4 Victorian Infectious Disease Reference Laboratory, Royal Melbourne Hospital at the Peter Doherty Institute for Infection and Immunity, Melbourne, Australia, 5 Department of Microbiology, Bayside Health, Melbourne, Australia, 6 Department of Infectious Diseases, University of Melbourne at the Peter Doherty Institute of Infection and Immunity, Melbourne, Australia, 7 Global HIV, Hepatitis and STI Programmes, World Health Organization, Geneva, Switzerland, 8 Independent Clinical Epidemiologist, Cape Town, South Africa, 9 University of California, Los Angeles, California, United States of America, 10 Joint United Nations Program on AIDS & HIV, Geneva, Switzerland, 11 Pan American Health Organization, World Health Organization, Geneva, Switzerland, 12 World Health Organization African Region, Geneva, Switzerland, 13 Faculty of Infectious and Tropical Diseases, London School of Hygiene and Tropical Medicine, London, United Kingdom

‡ These authors are joint first authors on this work.
* Jason.Ong@monash.edu

## Abstract

### Background

Long-acting injectable pre-exposure prophylaxis (LAI-PrEP) is a highly effective biomedical intervention for the prevention of HIV acquisition. There is a strong interest among communities and policymakers for LAI-PrEP scale-up, accelerating the demand for clear guidance on testing approaches that balance accuracy with scalability. Unlike oral pre-exposure prophylaxis, LAI-PrEP may overcome adherence challenges, such as difficulty with frequent clinic visits. However, LAI-PrEP results in prolonged subtherapeutic drug levels after discontinuation, which can increase the risk of drug resistance among those who have an undetected HIV infection. This systematic review evaluates how different HIV testing strategies, including rapid diagnostic tests (RDTs), laboratory-based immunoassays and nucleic acid testing (NAT), affect clinical utility and programme delivery of LAI-PrEP.

**Data availability statement:** All relevant data are within the manuscript and its Supporting information files.

**Funding:** This work was supported through resources provided to WHO (https://www.who.int) by external donors to support WHO guideline development activities. The external donors had no role in the study design, data collection and analysis, decision to publish, or preparation of the manuscript. CCJ, CBDS, MR, HLI, HAS, MP, BM, CL, HP, LC, PA, who are affiliated with and receive a salary from WHO, participated in data collection, data interpretation and writing of the report. JJO is funded by the WHO and the Australian National Health Medical Research Council Investigator Grant (GNT1193955) (https://www.nhmrc.gov.au).

**Competing interests:** I have read the journal's policy and the authors of this manuscript have the following competing interests: RJL serves as a consultant and serves on scientific advisory boards for ViiV and Merck. Other authors have declared that no competing interests exist.

**Abbreviations:** ART, antiretroviral therapy; CAB-LA, Cabotegravir Long-Acting; CDC, Centers for Disease Control and Prevention; CROI, Conference on Retroviruses and Opportunistic Infections; DVR, dapivirine vaginal ring; HIVST, HIV self-testing; IAS-USA, International Antiviral Society United States of America; INSTI, Integrase Strand Transfer Inhibitor; LAI-PrEP, long-acting injectable pre-exposure prophylaxis; LEN, lenacapavir; LMICs, low-and middle-income countries; NAT, nucleic acid testing; NPV, negative predictive value; OR, odds ratio; PPV, positive predictive value; PrEP, pre-exposure prophylaxis; PRISMA, Preferred Reporting Items for Systematic Reviews and Meta-Analyses; QUADAS-2, Quality Assessment of Diagnostic Accuracy Studies-2; RAMs, resistance-associated mutations; RDTs, rapid diagnostic tests; ROBINS-I V2, Risk Of Bias In Non-randomised Studies of Interventions, Version 2; WHO, World Health Organization.

## Methods and findings

We searched databases and retrieved studies up to April 8, 2025, and supplemented findings with data collected through a World Health Organization (WHO) survey among ongoing and completed LAI-PrEP implementation studies. We included publications reporting original or primary data on clinical, diagnostic and resource-use outcomes of HIV testing for LAI-PrEP. Meta-analyses were conducted using random-effects models. Chi-square tests were used to examine differences between related outcomes. Certainty of evidence was determined using the GRADE methodology (Prospero: CRD42024605562). Risk Of Bias In Non-randomised Studies of Interventions, Version 2 (ROBINS-I V2) assessment tool was used to assess bias for non-randomised comparative studies. Of 7,698 records identified, 38 reports representing 22 studies (cabotegravir: 20, lenacapavir: 2) across 15 countries were included. The overall certainty of evidence was low. Most were observational cohorts ($n = 13$) or non-randomised comparator studies ($n = 7$). Among 8,171 LAI-PrEP users in four randomised controlled trials, HIV detection rates were similar across strategies (9/8171 (RDT) versus 14/8171 (NAT) (Odds ratio (OR) 0.66 (95% confidence interval: 0.29–1.50; $P = 0.87$)), with no difference in adverse events. Compared with laboratory-based tests, RDTs enabled faster turnaround (same-day versus up to 7 days), more rapid treatment initiation (1 day versus 6–9 days), and lower test costs (US$4 versus US$22). All tests had similar negative predictive value (~100%) at LAI-PrEP initiation and comparable positive predictive value (~55%) at continuation. There was little difference in delayed HIV detection (11/8171 (RDT) versus 0/8171 (NAT)). In the HPTN 083 trial, NAT use was occasionally associated with false-positive results, leading to unnecessary PrEP holds or discontinuation (7/2483). NAT might have detected HIV before resistance emerged, though no prospective or modelling evidence showed clinical benefit at a population level. There was limited evidence of HIV self-testing for LAI-PrEP delivery. We noted that our assessment of performance accuracy in different testing strategies may introduce selection bias.

## Conclusions

RDT-based testing strategies have comparable accuracy to laboratory-based strategies and are more accessible and scalable, which can ensure that testing does not become a barrier to accessing or continuing LAI-PrEP. As countries expand access to LAI-PrEP amid increasingly constrained resources, adoption of new WHO guidance supporting the use of RDTs can enable simpler, more affordable, and user-centred HIV testing approaches.

PLOS Medicine

Author summary

## Why was this study done?

- Long-acting injectable pre-exposure prophylaxis (LAI-PrEP) is expanding rapidly, but it was unclear whether rapid diagnostic tests (RDTs), laboratory-based immunoassay tests (which detect HIV antibodies or antigen), or nucleic acid testing (NAT) (which detects HIV RNA) are the most appropriate HIV testing strategies for safe, effective and scalable delivery.

- Early or recently acquired HIV infections may not be detected by some testing strategies during LAI-PrEP because of lower viral loads, raising concerns about delayed diagnosis and the potential development of drug resistance.

- Policymakers require timely evidence on the accuracy, safety, and feasibility of HIV testing approaches to guide emerging global recommendations for LAI-PrEP rollout.

## What did the researchers do and find?

- We conducted a systematic review of HIV testing strategies, including RDTs, laboratory-based immunoassays, and NAT, and assessed their effects on clinical, diagnostic, and resource-use outcomes for LAI-PrEP delivery.

- Twenty-two studies from 15 countries were included, comprising 13 observational cohorts and 7 non-comparator studies.

- RDTs provided faster results, quicker treatment initiation, and lower testing costs, while NAT occasionally produced false-positive results that led to unnecessary delays or PrEP discontinuation.

- NAT might have identified HIV before resistance emerged in eight cases, but no prospective evidence demonstrated that NAT as part of real-time clinical decision-making would have prevented resistance.

- RDTs and laboratory-based immunoassay tests detected HIV slightly later than NAT, but this had no significant impact at the population level.

## What do these findings mean?

- RDT-based testing approaches appear safe, feasible, and practical for supporting LAI-PrEP scale-up, though the overall certainty of evidence is low and additional data are needed.

- NAT may detect a small number of infections earlier, but its routine use faces substantial feasibility challenges, and detecting one additional early infection would require testing more than 5,300 people at high cost.

- These findings suggest that RDTs can serve as the base testing strategy for LAI-PrEP programmes, while acknowledging the need for continued research, ongoing surveillance, and the selective use of NAT in specific clinical situations.

- We noted that our evaluation of the accuracy of different testing strategies may have introduced bias in the selection of participants.

## Introduction

HIV testing has been scaled up globally as a critical entry point into HIV care, treatment and prevention services, including pre-exposure prophylaxis (PrEP) [1]. The World Health Organization (WHO) recommends the use of HIV rapid diagnostic

tests (RDTs) as well as HIV self-testing (HIVST), a process in which individuals collect their own specimen, perform the test, and interpret the result themselves, to support equitable and wide-scale access to PrEP [2]. While this guidance has begun to support the roll-out of oral PrEP and the dapivirine vaginal ring (DVR), the applicability of these testing approaches for novel long-acting injectable PrEP (LAI-PrEP) options remains unclear [3].

LAI-PrEP is highly effective in preventing HIV acquisition and has been heralded as a scientific breakthrough for HIV prevention [4,5]. Compared with oral PrEP, LAI-PrEP may overcome adherence challenges, including forgetfulness, pill fatigue, and difficulty with frequent clinic visits [6]. Currently, there are two LAI-PrEP agents—cabotegravir (CAB-LA), administered every two months, and lenacapavir (LEN), given every six months—that have demonstrated superior efficacy over oral PrEP in randomised controlled trials [4,5,7,8].

Globally, the vast majority of all HIV testing is performed at the primary care and community level without clinical laboratories. Nearly all HIV tests used in low- and middle-income countries (LMICs) are RDTs. HIV RDTs have been widely used since the 1990s, and HIV self-testing (HIVST) use has been growing for the last decade, as both can facilitate accurate diagnosis and prompt access to treatment and prevention services [9]. WHO has prequalified more than 20 HIV RDTs and 7 HIVST kits, all of which are of high quality, accurate and affordable in resource-limited settings.

LAI-PrEP, however, may introduce distinct clinical considerations for HIV testing services [10]. A key concern is the potential initiation of LAI-PrEP in individuals with undiagnosed HIV infection, which may result in functional monotherapy and increase the risk of developing drug resistance. Unlike oral PrEP, where HIV protection ceases shortly after stopping medication, LAI-PrEP results in prolonged drug exposure, with subtherapeutic concentrations persisting for weeks to months following the last injection [11]. To address this concern, early policies, including those from WHO, were uncertain whether testing using HIV RDTs and HIVST alone would be sufficient for public health programmes to deliver LAI-PrEP. Current policies from the U.S. Centers for Disease Control and Prevention (CDC) and the International Antiviral Society United States of America (IAS-USA) continue to advise laboratory-based immunoassay testing and the use of nucleic acid testing (NAT) or RNA testing as part of LAI-PrEP delivery [12,13].

There is a strong interest among communities and policymakers for LAI PrEP scale-up, accelerating the demand for clear guidance on testing approaches that balance accuracy with scalability [14]. Given the pace at which LAI-PrEP is being introduced, robust data is needed for the development of provisional, pragmatic testing recommendations. This review evaluates how different testing approaches, including RDTs, HIVST, laboratory-based immunoassays and NAT techniques, affect clinical outcomes and programme delivery for LAI-PrEP. Diagnostic accuracy is important, but the clinical and public health value of LAI-PrEP depends equally on whether testing strategies are feasible, scalable, and accessible. Our systematic review examined clinical and diagnostic outcomes across LAI-PrEP initiation, re-initiation and continuation to inform WHO guidelines.

## Methods

This systematic review was conducted following WHO Handbook for Guideline Development [15] and the Cochrane Handbook for Systematic Reviews of Interventions [16]. This study is reported as per the Preferred Reporting Items for Systematic Reviews and Meta-Analyses (PRISMA) guideline (S1 Checklist) [17]. Our review protocol was registered on Prospero (CRD42024605562).

### Search strategy and selection criteria

We searched OVID Medline, OVID Embase, CINAHL, and Cochrane Central Register of Controlled Trials for studies published from inception to October 18, 2024. We also searched electronic conference abstracts from the International AIDS Society (AIDS), the Conference on Retroviruses and Opportunistic Infections (CROI), and the HIV Research for Prevention (HIVR4P) for all available abstracts. Additionally, relevant studies were retrieved from WHO research team and WHO open call. This resulted in the inclusion of studies published up to 8 April 2025. We also contacted the authors to retrieve

relevant additional information. No restrictions were placed on the search. The search strategy protocol can be found in S1 Appendix. Search results were imported into Covidence, where duplicate studies were excluded. Two independent researchers (WT, EW and JJO) screened each title and abstract. Two independent researchers (WT and EW) screened full-text articles and extracted data into Excel sheets. Discrepancies were resolved by a third researcher (JJO) and discussion among authors (WT, EW and JJO).

We included publications reporting original or primary data on clinical outcomes and diagnostic outcomes of HIV testing for LAI-PrEP, including CAB-LA and LEN. HIV testing included NAT, laboratory-based immunoassays, RDT (incorporating point-of-care testing) and self-testing. Definitions of the clinical and diagnostic outcomes are provided in S2 Appendix. Although we used evidence from randomised controlled trials, we classified them in our review as non-randomised comparator studies because the point of randomisation was not for the testing strategy. Indirect evidence (e.g., data related to broader PrEP delivery) was used when direct evidence was unavailable.

In accordance with WHO policy and GRADE methodology, members of the WHO guideline development group ranked outcomes a priori based on their level of importance. The following outcomes were ranked as critical: time to linkage and antiretroviral therapy (ART) initiation, PrEP holds or discontinuations from false positives, HIV positivity, diagnostic accuracy and performance, time of detection, resistance-associated mutations (RAMs), and turnaround time of test results. The remaining outcomes were ranked as important: testing frequency, sexual risk behaviour and clinical/social harms.

## Data analysis

Descriptive statistics were used to summarise PrEP holds or discontinuations from false positives, turnaround time of test results, testing frequency, RAMs at first evidence of HIV and clinical or social harms. We used the Chi-square test to examine the difference in HIV positivity and delays in detection. Meta-analyses were conducted using a random-effects model to calculate odds ratios (OR) with corresponding 95% confidence intervals for studies reporting comparable outcomes. We reported positive predictive value (PPV) and negative predictive value (NPV) for diagnostic accuracy outcomes. Although the protocol prespecified a bivariate analysis of diagnostic accuracy, the necessary paired $2 \times 2$ data were not available across studies. Due to inconsistent testing algorithms and retrospective use of NAT in several trials, a formal bivariate meta-analysis could not be performed. We reported the absolute risk of delayed HIV detection at the population/programmatic level to compare across testing strategies in terms of absolute numbers of delayed diagnoses per 1,000 people tested, which is the relevant metric for implementation planning. All statistical analyses were performed using Stata BE 18.0 (StataCorp. 2023. *Stata Statistical Software: Release 18*. College Station, TX: StataCorp LLC.) or Review Manager (RevMan, version 5.4, The Cochrane Collaboration, 2020). Additional information on values and preferences, feasibility, equity and resource use were collected and summarised descriptively.

## Quality assessment

Risk Of Bias In Non-randomised Studies of Interventions, Version 2 (ROBINS-I V2) assessment tool was used to assess bias for non-randomised comparative studies [18], and Quality Assessment of Diagnostic Accuracy Studies-2 (QUADAS-2) was used to assess bias for diagnostic accuracy studies (S3 Appendix) [19]. GRADE methodology was used to assess the certainty of evidence (S4 Appendix) [15,20,21].

## Financial disclosure statement

This work was supported through resources provided to WHO (https://www.who.int) by external donors to support WHO guideline development activities. The external donors had no role in the study design, data collection and analysis, decision to publish, or preparation of the manuscript. CCJ, CBDS, MR, HLI, HAS, MP, BM, CL, HP, LC, PA, who are affiliated with and

receive a salary from WHO, participated in data collection, data interpretation and writing of the report. JJO is funded by WHO and the Australian National Health Medical Research Council Investigator Grant (GNT1193955) (https://www.nhmrc.gov.au).

## Results

### Study selection and characteristics

Of 7,698 unique records identified, 86 records from the database searches and 30 from other sources (WHO survey, which collated data up to 1st December 2024; conferences; and author correspondence) were screened for eligibility. Overall, 38 records representing 22 studies (i.e., 20 studies for CAB-LA and 2 for LEN) reporting clinical and diagnostic outcomes and 6 studies reporting resource use [22–27] met the eligibility criteria and were included (Fig 1). Table 1 and S5 Appendix summarise the details of the included studies. The overall certainty of evidence was low (S4 Appendix and Table 2).

### Benefits of using an RDT-based strategy compared with NAT

Compared with a NAT-based strategy, RDT may lead to faster ART initiations after a positive HIV diagnosis (S6 Appendix) *(Very low certainty).* Two studies from the United States (US) (SeroPrEP Case Series and Case Report) [28,29] reported that two individuals who used NAT and laboratory-based immunoassays took 6 and 9 days to initiate ART post-diagnosis, respectively. In an observational cohort in Zimbabwe [30], using an HIV RDT it took one individual one day to initiate ART post-diagnosis [30].

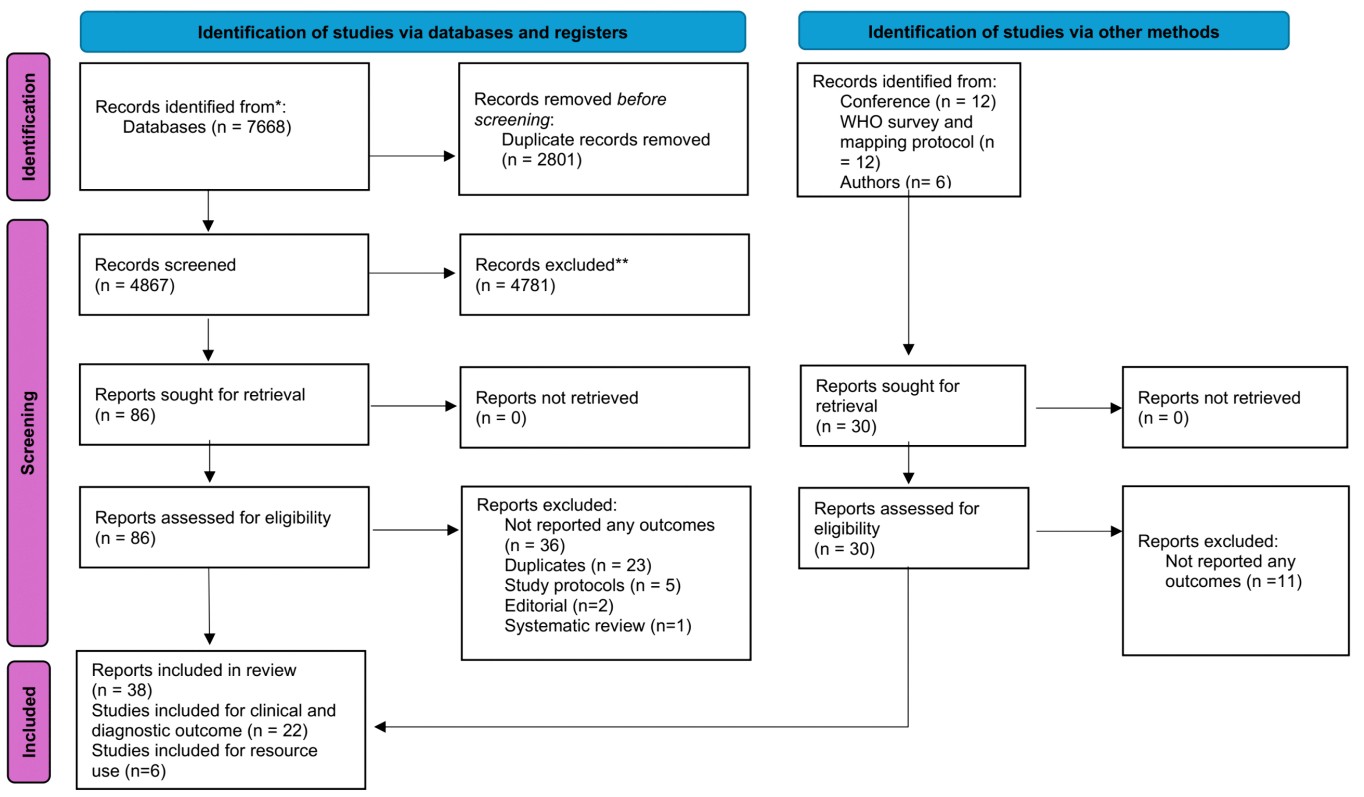

**Fig 1. PRISMA flow diagram.**

**Table 1. Summary of the main characteristics of the included studies (N = 22).**

| | n | % |
|---|---|---|
| **Type of study (according to HIV test strategy)*** | | |
| Non-randomised comparator studies | 7 | 32 |
| Observational cohort | 13 | 59 |
| Case report | 1 | 5 |
| Case series | 1 | 5 |
| **Population type¥** | | |
| Non-specific | 5 | 23 |
| Men who have sex with men and/or trans and gender diverse | 4 | 18 |
| Non-binary/sex diverse | 2 | 9 |
| Women | 4 | 18 |
| Men who travel for work | 1 | 5 |
| Pregnancy, infant and maternal health | 3 | 14 |
| Young people | 2 | 9 |
| Not reported | 4 | 18 |
| **Type of long-acting injectable pre-exposure prophylaxis** | | |
| Cabotegravir | 20 | 91 |
| Lenacapavir | 2 | 9 |
| **Country** | | |
| United States of America | 4 | 18 |
| Brazil | 2 | 9 |
| Multiple countries in Africa | 13 | 59 |
| Multiple countries (global) | 2 | 9 |
| Not reported | 1 | 5 |

Detailed study-level characteristics can be found in S5 Appendix.

*Type of study was reported according to the outcomes that we included in this review instead of the type of study reported in the publication.

¥Some studies included more than one population. We used 22 as the denominator.

NAT-based strategies may result in PrEP holds or discontinuation from false positives (S7 Appendix), but the evidence is uncertain *(Very low certainty)*. This was based on data from the HPTN 083 trial, where 22/2483 (0.9%) individuals received a false positive RNA test, resulting in four who delayed CAB-LA initiation, two who discontinued CAB-LA and one who had delayed oral CAB; none of these seven individuals had evidence of HIV [31]. In total, 20/2483 (0.8%) of individuals received a false positive RDT (unspecified) and/or laboratory-based immunoassay test, but no further data were available on whether this resulted in PrEP holds or discontinuation [31]. The CATALYST study reported 2/1010 (0.2%) individuals with a false positive NAT test. However, both continued CAB-LA [30].

Compared to NAT or laboratory-based immunoassay strategies, RDT-based strategies appear to make little or no difference in detection of HIV positivity while using CAB-LA or LEN (S8 Appendix) *(Low certainty)*. For CAB-LA and LEN, 14/8171 (0.2%) acquired HIV [4,5,7,8]. There was no difference in HIV positivity comparing all three testing strategies ($p = 0.56$), RDT versus NAT ($p = 0.40$) or RDT versus laboratory-based immunoassay ($p = 0.52$). RDTs may make little to no difference in the detection of new HIV infections (9/8171) compared to NAT-based strategies (14/8171), with an OR of 0.66 (95% CI: 0.29–1.50; $P = 0.87$) and an absolute difference of 1 fewer infection per 1,000 tests (range: 1 fewer to 1 more).

**Table 2. Summary of findings.**

| Outcome | Findings | Certainty |
|---|---|---|
| **Clinical outcomes** | | |
| Time to ART initiation after diagnosis | RDT may lead to faster ART initiation (e.g., 1 day in Zimbabwe) compared to NAT and laboratory-based immunoassay testing (6–9 days in the US) (S6 Appendix) | Very low |
| PrEP holds and discontinuation from false positives | NAT may lead to PrEP holds or discontinuation due to false positives (e.g., 7/2483 affected in HPTN 083). No data available for RDT and laboratory-based immunoassay testing. (S7 Appendix) | Very low |
| HIV positivity rates | There may be no significant difference in the detection of HIV positivity between RDT, NAT, and laboratory-based immunoassays (OR 0.66 [0.29–1.50]; p = 0.40). (S8 Appendix) | Low |
| Delayed HIV detection | RDT and laboratory-based immunoassays may be more likely than NAT to delay detection (OR 7.08 [1.87–26.87]), but absolute differences per 1,000 users were negligible (11/8171 for RDT vs. 0/8171 for NAT). (Fig 2) | Low |
| Testing frequency | There may be no difference in testing frequency across strategies. | Low |
| Social harms | No data available on social harms from misdiagnosis. | Low |
| **Diagnostic outcomes** | | |
| Diagnostic accuracy | Sensitivity and PPV at CAB-LA continuation: RDT/laboratory-based immunoassay testing may have lower sensitivity (82.8%) than NAT (100%). PPVs were similar (~55%). NPV at initiation: RDT, NAT, and laboratory-based immunoassay testing all showed NPV ~100% for both CAB and LEN. (S9 Appendix) | Low |
| Turnaround time for results | RDT results are available at the same consultation; laboratory-based immunoassay tests typically take 1–5 days, and NAT takes up to a week. (S10 Appendix) | Very low |
| Resistance-associated mutations | INSTI RAMS were detected in 0.3% of CAB and capsid RAMS detected in 0.05% of LEN users. No direct link to testing strategy. RNA testing might detect HIV before INSTIS RAM emerged and enable earlier ART in 0.09% (8/2282) CAB users. (S11 Appendix) | Low |
| **Resource use** | | |
| Cost and resource use | RDTS are significantly cheaper than NAT ($4 vs. $22 USD/test). To detect 1 additional HIV case, missed by a rapid test, using NAT requires testing 5,305 people, with estimated costs $46,684 - $451,456. | Low |

GRADE methodology was used to assess the certainty of evidence (S4 Appendix).

ART,antiretroviral therapy; CAB-LA, long-acting cabotegravir; HPTN, HIV Prevention Trials Network; INSTI,integrase inhibitor; HIVST, HIV self-testing; LEN, Lenacapavir; NAT, nucleic acid test; NPV, negative predictive value; OR, odd ratio; PPV, positive predictive value; PrEP, pre-exposure prophylaxis; RAM, resistance-associated mutation; RDT, rapid diagnostic test; US, United States; USD, United States dollar.

Compared to NAT, RDT-based strategy or laboratory-based immunoassay strategies may make little to no difference in NPV at initiation (S9 Appendix) *(Low certainty).* Comparing RDT to NAT: for CAB, [4,7,30,32–41] NPV was 100% (99.9%–100%), and for LEN [5,8] NPV was 100% (99.9%–100%). Comparing laboratory-based immunoassay testing to NAT: for CAB [4,7,30,32–35,37–42] NPV was 100% (99.9%–100%), and for LEN [5,8] NPV was 100% (99.9%–100%).

Compared to NAT, RDT and laboratory-based immunoassay strategies may have reduced sensitivity but achieve similar PPV for CAB-LA continuation (S9 Appendix) *(Low certainty).* This is based on data from HPTN 083 open-label extension [43] where there was a sensitivity of 82.8% (65.5%–92.4%) and PPV of 54.6% (40.1–68.3) for detecting HIV if using RDT and laboratory-based immunoassay, compared with sensitivity of 100% (88.3–100) and PPV of 56.9% (43.3–69.5)

when using NAT. If using NAT within six months of the CAB-LA injection, sensitivity was 87.5% (46.7–99.3), and PPV was 9.1% (1.6–30.6). However, if the CAB injection was after 6 months, the sensitivity was 100% (80–100), and PPV was 60% (17%–92.7%).

Compared to NAT, RDTs may have little to no effect on the absolute numbers of delayed detection of HIV for CAB-LA or LEN users at initiation (Figs 2 and 3) *(Low certainty)*. For CAB-LA, 5/3858 (0.1%) had delayed detection compared with NAT [4,7,28,29,32–35,37–39,41,42,44] (median 63 days (interquartile range: 46–91.5). For LEN, 8/4313 (0.2%) had delayed detection (5 weeks for one case) [5,8]. RDT is significantly more likely than NAT to have delayed HIV detection, with an OR of 7.08 (1.87–26.87) (Fig 2), but this translates to no absolute difference in delayed detections per 1,000 tests (range: 0 fewer to 0 fewer).

Compared with laboratory-based immunoassays or NAT, RDT-based strategies may have a shorter turnaround time (S10 Appendix) *(Very low certainty)*. RDT results are available at the same consultation [30], whereas laboratory-based immunoassay results are usually available within 1–5 days, and qualitative NAT results can take a week [5,8].

Compared to NAT, an RDT-based strategy may have no difference in testing frequency *(Low certainty)*. Based on 10 studies, NAT, RDTs and/or laboratory-based immunoassays strategies may not change the testing frequency for LAI-PrEP users [4,5,7,8,30]. In a few studies, the use of HIVST was recommended 24 hours before CAB-LA injection (ImPrEP CAB Brasil [30]) and used two weeks after CAB-LA initiation in men who travel for work (MOBILE MEN) [30].

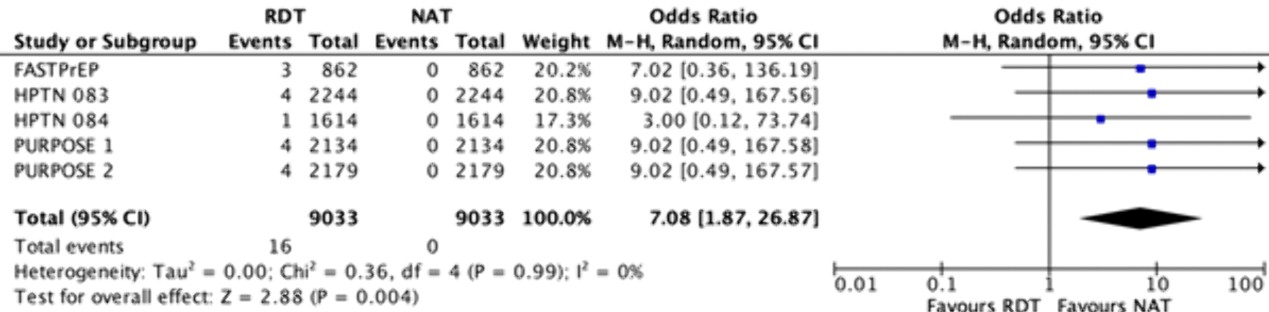

**Fig 2. Forest plot of the number of delayed detections of new HIV cases, comparing rapid diagnostic tests with nucleic acid tests when initiating injectable LAI-PrEP.** Cochran's chi-squared statistic for heterogeneity; CI, Confidence interval; df, Degrees of freedom; $I^2$, Higgins' $I$-squared statistic for heterogeneity; LAI-PrEP, Long-acting injectable pre-exposure prophylaxis; M-H, Mantel–Haenszel method; NAT, Nucleic acid test; P, $P$-value; RDT, Rapid diagnostic test; Tau$^2$, Tau-squared; Z, Z-statistic for the overall pooled effect.

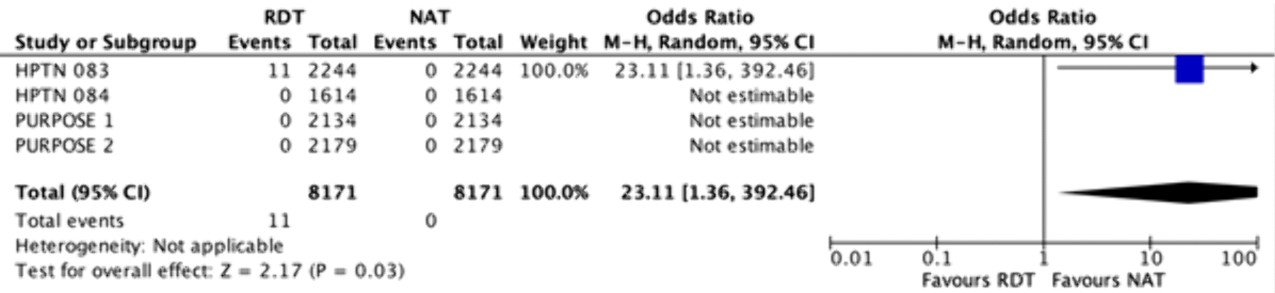

**Fig 3. Forest plot of the number of delayed detections of new HIV cases, comparing rapid diagnostic tests with nucleic acid tests when continuing injectable LAI-PrEP.** Cochran's chi-squared statistic for heterogeneity; CI, Confidence interval; df, Degrees of freedom; $I^2$, Higgins' $I$-squared statistic for heterogeneity; LAI-PrEP, Long-acting injectable pre-exposure prophylaxis; M-H, Mantel–Haenszel method; NAT, Nucleic acid test; P, $P$-value; RDT, Rapid diagnostic test; Tau$^2$, Tau-squared; Z, Z-statistic for the overall pooled effect.

## Undesirable effects of using RDT

Compared to NAT, an RDT-based strategy may not cause clinical or social harms *(Low certainty).* No data regarding social harms (including adverse events from misdiagnosis, anxiety, and domestic violence from false positives) or sexual risk behaviours of those with misdiagnosis were available. Seven studies reported no social harms among LEN and CAB-LA users (not specific to misdiagnosis) [4,5,7,8,28,30,32–35,37–39,41,42,44,45].

There may be detection of RAMs at first evidence of HIV at the individual level for CAB and LEN users (S11 Appendix) *(Low certainty).* However, there is no direct evidence that the testing strategy would affect the development of RAMs. Major integrase inhibitor (INSTI) RAMs were detected in 10/3858 (0.3%) CAB users compared with TDF/FTC 0/3858 (0%, p-value = 0.002) [4,7,28,32–35,37–39,41,42,44]. Of the 10 major INSTI RAMs detected in CAB users described in HPTN 083, retrospective testing of stored samples after a RDT positive result suggests that NAT could have detected the HIV infection before INSTI RAMs emerged in eight cases [4,7,28,32–35,37–39,41,42,44]. However, this has not been demonstrated in prospective studies. Data from PURPOSE 1 and PURPOSE 2 reported capsid RAM in 2/4313 (0.05%) of LEN users, 0/2154 (0%) of TDF/FTC users and 0/2136 (0%) of TAF/FTC users [5,8].

## Resource use

Compared to NAT, the cost of implementing RDT-based testing strategies may be lower *(Low certainty).* A modelling study of CAB-LA in sub-Saharan Africa reported that NAT commodities are much more expensive ($22 United States Dollars (USD)) compared with 3rd generation RDT ($4 USD), with a small benefit in reducing resistance but no improvement in survival or overall treatment success [27]. Other studies confirm that the cost of NAT is significantly more than that of RDT or laboratory-based immunoassays (S12 Appendix) [22–26]. We calculated, based on the reported data, that to detect one additional HIV case, missed by a rapid test, using NAT requires testing 5,305 people with estimated costs of $46,684–$451,456 [42].

Indirect evidence from a modelling study of oral PrEP reported that the costs for PrEP provision were similar for HIVST and RDT, but PrEP scale-up using NAT was approximately 50% more expensive, mainly due to higher testing costs [26].

## Acceptability and impact on equity

RDTs (compared to laboratory-based testing) could improve equitable access to LAI-PrEP by expanding service delivery options, especially for underserved populations. Limited NAT testing capacity in low-and LMICs presents a barrier to LAI-PrEP implementation, with high costs and logistical demands limiting scalability. In HPTN 083, 11 of 22 false-positive NAT results originated from an LMIC site. A US study found 30% of PrEP users received a new PrEP prescription without a laboratory-based HIV immunoassay test, and 25% without any HIV test in the prior three months, indicating challenges in meeting testing requirements at scale, even in a high-income country setting [46].

## Discussion

This systematic review synthesises evidence on different LAI-PrEP testing strategies, focussing on RDTs versus NAT or laboratory-based immunoassays. A key contribution of this work is its focus on the clinical utility and program delivery implications of testing strategies, rather than diagnostic accuracy in isolation. We found similarities in outcomes between RDT, laboratory-based immunoassay and NAT-based testing strategies, including HIV positivity rate, NPV, delays in detection, testing frequency, and no clinical or social harms. While NAT offers benefits in higher diagnostic sensitivity, and may have detected some INSTI RAMs before they emerged, RDTs provide several advantages for patients' outcomes and management, including quicker results, faster linkage to care and ART initiation, lower costs, and higher feasibility and accessibility. Further evidence on implementing HIVST to enhance access to LAI-PrEP is needed.

Integrating RDT-based testing into LAI-PrEP programs offers some advantages over NAT and laboratory-based immunoassays. Testing services are essential for safe PrEP delivery, streamlining service provision and supporting public health approaches. To ensure effective and equitable PrEP access, HIV testing must be feasible, accessible, and scalable. Our findings showed that RDT provided quicker results than NAT. While laboratory-based test results are usually returned within a week, programme-level turnaround times can vary substantially across settings. For example, in early infant diagnosis programmes, the median time from sample collection to result delivery to the caregiver has been reported as long as 35 days and 60 days, compared with same-day results for RDTs [27,47]. This difference could be from operational and communication delays rather than laboratory processing time, and it highlights that RDTs are more likely to consistently provide substantially faster results, linkage to care and ART initiation across diverse service settings—a pattern likely to be similar when implementing RDTs within LAI-PrEP delivery

Simplified and affordable testing approaches that rely on RDTs and HIVST are important to ensuring LAI-PrEP has a public health impact. Our findings highlight the benefits of transitioning global programming to RDTs, particularly for marginalised and mobile populations or those in LMICs because of their ease of use, reduced infrastructure requirements and widespread availability [48]. In many settings, the use of an RDT as the initial screening test supports same-day PrEP initiation, consistent with WHO recommendations aimed at improving uptake and adherence among key populations. Importantly, LAI-PrEP may be prioritised for use among individuals for whom existing HIV prevention options—including daily oral PrEP—have not been acceptable, feasible, or desirable. For these populations, rigid or complex testing requirements risk creating additional barriers to access. HIV testing strategies for LAI-PrEP should therefore prioritise flexibility and user-centred delivery models that can accommodate diverse service delivery contexts while maintaining safe and effective prevention. Given this, guidance that requires confirmatory laboratory-based testing prior to, or for continuing, LAI-PrEP initiation warrants careful consideration. Whilst confirmatory testing has clear diagnostic value, particularly for detecting early infection, making it a mandatory prerequisite may have unintended consequences leading to unnecessary rigidity to service delivery, as well as delays and limited access and uptake in critical populations and resource-constrained settings. This issue is evident in the US. Despite CDC recommendations, only 35% of those who initiated CAB-LA tested by HIV RNA within ±14 days at initiation [49]. Although missed early diagnoses may have broader economic implications—including the cost of PrEP provision, delayed diagnosis, potential resistance, and onward transmission—the included studies did not provide sufficient prospective data to quantify these downstream effects. A modelling study shows that NAT adds significant cost to CAB-LA rollout programs in LMICs, but provides a small benefit in reducing resistance, and does not improve survival or overall treatment success [27]. Even high-income countries will benefit from RDT-based testing strategies, as evidence shows substantial feasibility constraints of using NAT for routine programmatic implementation [46]. This does not imply that NAT has no role in LAI-PrEP delivery. There may be settings and situations where it could be considered based on available resources. While individuals and programmes may use various PrEP options, such as oral PrEP and DVRs, using RDT-based testing strategies provides a practical and scalable alternative that supports differentiated service delivery for PrEP, ensuring timely access to PrEP continuation and linkage to confirmatory testing, particularly in resource-limited settings. Considering the growing financial constraints facing global health, and especially HIV testing and prevention programmes, simple and flexible tools are essential for future LAI-PrEP delivery.

RDTs have reduced sensitivity in detecting HIV in its early stages [50], but our analysis suggests this has minimal impact on absolute clinical outcomes at the population level. Despite some difference in the expected window periods of various HIV detection methods, missed infections may be picked up at next clinical visits and be linked to effective treatment. Historically, delayed diagnosis has been a concern due to the high viral loads observed in early infection in PrEP-naive individuals [51,52]. Given the lower viral loads observed in early LAI-PrEP infections, the transmission risk associated with these infections is likely to be lower than for PrEP-naive individuals [53]. Our findings showed that NAT could theoretically detect early or rare breakthrough infections during LAI-PrEP use. However, because NAT was used retrospectively on stored samples in HPTN 083 and 084—not prospectively as part of real-time clinical

decision-making—there is no prospective evidence that earlier NAT detection would have prevented the development of INSTI RAMs. In the real world, high-frequency follow-up testing, as done in studies, will likely be infeasible, and turn-around times in resource-limited settings may be lengthy. As a result, the benefit of a slightly earlier diagnosis may be limited and ultimately not affect the small number of cases with resistance [27]. Based on the data from real-world implementation, despite lower sensitivity of RDTs, there is little to no absolute difference in delayed HIV detection per 1,000 tests when using RDTs among CAB-LA or LEN users at initiation and for CAB-LA users at continuation. Further, using data from HPTN 083 open-label extension [42], detecting one additional case using NAT—missed by RDT at an earlier time point—would require testing 5,305 people using NAT. This suggests that a lot of resources are needed for minimal gain, using a public health lens.

Another concern regarding the use of RDTs compared to NAT-based testing strategies for LAI-PrEP is the potential for delayed diagnosis to contribute to the development of major RAMs [54], particularly those with cross-resistance to current first-line INSTI therapies. However, it is important to note the limitations of these data. First, there is no direct evidence that NAT-based testing would prevent the development of resistance. Second, due to low viral loads associated with early infection, several of the cases with major INSTI RAMs, were detected using a single-genome sequencing INSTI drug resistance assay [55,56]. This research methodology is usually not available for use in clinical diagnostic laboratories and has not been validated in the clinical diagnostic setting. Third, from indirect evidence, the use of HIVST compared with NAT or laboratory-based immunoassays had no significant impact on HIV deaths averted, despite a potential rise in resistance [26]. Fourth, there have been no well-described ART failures due to LAI-PrEP-associated INSTI resistance. Further, additional modelling suggests that the total number of resistance cases remains small even in very high incidence groups, such as MSM in Thailand [57]. While more sensitive tests could reduce resistance, the impact is minimal [27,57]. For example, at 100% uptake of CAB-LA with RDTs, 6,200 new infections could be averted, and the difference in the number of INSTI-resistant and transmitted resistance cases remains very low (INSTI resistance: RDT: 94 versus NAT: 36, transmitted resistance: RDT: 15 versus NAT: 6). Thus, while more sensitive HIV testing will prevent some INSTI resistance, given the low INSTI resistance incidence, more sensitive testing is not critical for CAB-LA rollout [27,57]. The benefits of a programme that is able to scale and get more people on PrEP, and at a fraction of the cost and more access-driven approaches, appear beneficial. Scaling-up LAI-PrEP requires not only simplified HIV testing but also an investment in monitoring for LAI-PrEP-associated resistance within established HIV surveillance programs. This should include monitoring for both LAI-PrEP-associated major RAMs and LAI-PrEP-associated treatment failures.

We identified limited evidence on HIVST for LAI-PrEP delivery. Outcomes data on HIVST were only available for testing frequency. Novel uses of HIVST for LAI-PrEP delivery are emerging, such as in Brazil, where clients self-test before their first LAI-PrEP [1,10]. Emerging evidence suggests HIVST, performing similarly to HIV RDTs, might be sufficiently accurate for LAI-PrEP implementation [30]. While this shows promise, and HIVST can be an option, we noted that oral-fluid HIVST or RDTs have a considerably longer window period than blood-based RDTs and laboratory assays [58,59]. Further implementation research is needed to fully understand the role of HIVST in guiding LAI-PrEP delivery. Given evolving technology, low-cost blood-based HIVST or other innovative self-collection tools should be considered a priority for investigation.

Several gaps require further investigation to optimise HIV testing strategies for LAI-PrEP. To date, no prospective studies have directly compared the clinical or diagnostic outcomes of different testing strategies for LAI-PrEP. We acknowledge the limited number of studies, heterogeneity in study methods, and risk-of-bias concerns. Consequently, the overall certainty of evidence remains very low or low and could be strengthened through further modelling analyses and prospective studies that provide insight into broader economic implications (e.g., the cost of PrEP provision, delayed diagnosis, potential resistance, and onward transmission), optimal testing strategies and approaches for LAI-PrEP, including for new and emerging PrEP agents. Flexible and scalable HIV testing approaches are crucial to prevent testing from hindering access to long-acting injectable options. At the same time, it remains important for programmes to invest in low-cost drug resistance monitoring and surveillance, especially as LAI-PrEP becomes more widely available. Encouragingly, several

countries in resource-limited settings are moving forward with rapid tests and evaluating self-tests. While findings are encouraging, further monitoring and analysis will be critical to guiding programme scale-up.

Our study's main strength was that we systematically reviewed all published papers related to HIV testing in LAI-PrEP, collaborating with WHO for information through an open call, and directly contacting authors for details. However, our assessment of performance accuracy in different testing strategies may introduce selection bias. For example, studies (e.g., HPTN 083 and 084) included in this review used NAT for pre-screening and RDT/laboratory-based immunoassay at enrolment, potentially favouring NAT's sensitivity estimates, as we could only compare new seroconversions at the initiation and continuation of CAB-LA. The gap between pre-screening and enrolment also created the potential for undetected seroconversion. Individuals with a positive RDT/laboratory-based immunoassay/NAT were excluded from enrolment, whereas NAT was retrospectively performed on stored samples post-enrolment. This screening strategy likely explains the zero infections detected by RDT/laboratory-based immunoassay at CAB-LA initiation, compared to 13 by NAT. Importantly, there is no direct evidence confirming that pre-enrolment NAT results were not false negatives, nor can the absolute number of false negatives from NAT be determined, as NAT was only performed when RDT was positive. Additionally, there were limited data on the clinical and diagnostic outcomes for HIVST, social harms and sexual behaviours of those receiving misdiagnosis or developing RAMs from misdiagnosis. Due to these limitations, our findings and conclusions about the comparative performance of different testing strategies for LAI-PrEP should be interpreted cautiously. Lastly, prior to final submission of this paper, we identified seven additional publications [60–66]. Four were full-text reports of studies already included in this review [61–64], two reported behavioural and psychosocial factors among LAI-PrEP users [60,65] and one abstract about the impact of rapid HIV testing and self-testing on LEN effectiveness and resistance [66].

In conclusion, delayed HIV diagnosis and breakthrough infections were rare and absolute differences in detection between different HIV testing strategies were minimal. Although NAT may offer high sensitivity and early detection—and remains important in selected cases—this did not translate into meaningful clinical or programmatic improvements for LAI-PrEP delivery. In contrast, simpler RDT-based strategies were low-cost, scalable and feasible to implement across a range of settings. As LAI-PrEP continues to scale globally, testing approaches should prioritise flexibility, accessibility, and affordability—factors that are critical for achieving population-level HIV prevention impact.

## Supporting information

**S1 Appendix. Search strategy protocol.**
(DOCX)

**S2 Appendix. Definition of outcomes.**
(DOCX)

**S3 Appendix. Risk of bias assessment.**
(DOCX)

**S4 Appendix. Certainty of evidence (GRADE).**
(DOCX)

**S5 Appendix. Characteristics and testing strategies of each program.**
(DOCX)

**S6 Appendix. Timlinkage to care.**
(DOCX)

**S7 Appendix. PrEP holds or discontinuation from false positive.**
(DOCX)

**S8 Appendix. HIV positivity.**
(DOCX)

**S9 Appendix. Diagnostic accuracy and performance.**
(DOCX)

**S10 Appendix. Turnaround time for results.**
(DOCX)

**S11 Appendix. Resistance-associated mutations.**
(DOCX)

**S12 Appendix. Costs of HIV tests (USD 2024).**
(DOCX)

**S1 Checklist. PRISMA 2020 Checklist and PRISMA 2020 for Abstract Checklist.** From: Page MJ, McKenzie JE, Bossuyt PM, Boutron I, Hoffmann TC, Mulrow CD, and colleagues. The PRISMA 2020 statement: an updated guideline for reporting systematic reviews. BMJ 2021;372:n71. https://doi.org/10.1136/bmj.n71. This work is licensed under CC BY 4.0. (https://creativecommons.org/licenses/by/4.0/).
(DOCX)

## Acknowledgments

We acknowledge the participants of WHO open call for their contribution of data.

During the preparation of this work, the authors used Grammarly AI in order to improve language and readability. After using this tool/service, the authors reviewed and edited the content as needed and took full responsibility for the content of the publication.

## Author contributions

**Conceptualisation:** Cheryl C. Johnson, Jason J. Ong.

**Data curation:** Warittha Tieosapjaroen, Eloise Williams, Jason J. Ong.

**Formal analysis:** Warittha Tieosapjaroen, Eloise Williams, Jason J. Ong.

**Funding acquisition:** Cheryl C. Johnson, Jason J. Ong.

**Methodology:** Warittha Tieosapjaroen, Eloise Williams.

**Supervision:** Jason J. Ong.

**Validation:** Nandi Siegfried, Jason J. Ong.

**Writing – original draft:** Warittha Tieosapjaroen, Eloise Williams, Jason J. Ong.

**Writing – review & editing:** Warittha Tieosapjaroen, Eloise Williams, Cheryl C. Johnson, Carlota Baptista Da Silva, Magdalena Barr-DiChiara, Michelle Rodolph, Heather Leigh Ingold, Heather-Marie A. Schmidt, Mateo Prochazka, Busi Msimanga, Celine Lastrucci, Hortensia Peralta, Lastone Chitembo, Precious Andifasi, Nandi Siegfried, Raphael J. Landovitz, Jason J. Ong.

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
