## [Editor Report · Decision Letter 0]

10 Jun 2025

Dear Dr Ong,

Thank you for submitting your manuscript entitled "Balancing Accuracy, Accessibility, and Cost: HIV Testing Strategies for Long-Acting Injectable Pre-Exposure Prophylaxis: A Systematic Review of Rapid, Laboratory-Based Immunoassay and Nucleic Acid Testing Strategies" for consideration by PLOS Medicine.

Your manuscript has now been evaluated by the PLOS Medicine editorial staff as well as by an academic editor with relevant expertise and I am writing to let you know that we would like to send your submission out for external peer review.

For clinical studies, please upload a copy of your trial study protocol as a supporting information file. The study protocol should be the version submitted for approval to the institutional review board or ethics committee, should include any amendments to the study protocol, as well as the date of their approval by the institutional review or ethics committee. Please also detail any deviations from the study protocol in the Methods section of your manuscript. The editors will consider the protocol and study conduct prior to a final decision for external review.

Please re-submit your manuscript within two working days, i.e. by Jun 12 2025 11:59PM.

Kind regards,

Alison Farrell, Ph.D.

Senior Editor

PLOS Medicine

---

## [Decision Letter · Decision Letter 1]

21 Nov 2025

Dear Dr Ong,

Many thanks for submitting your manuscript "Balancing Accuracy, Accessibility, and Cost: HIV Testing Strategies for Long-Acting Injectable Pre-Exposure Prophylaxis: A Systematic Review of Rapid, Laboratory-Based Immunoassay and Nucleic Acid Testing Strategies" (PMEDICINE-D-25-02056R1) to PLOS Medicine. I apologize for the delay in conveying to you our decision as we have been waiting for additional input on the study. The paper has now been reviewed by subject experts and a statistician; their comments are included below and can also be accessed here: [LINK]

As you will see, while the reviewers find the topic important, they require a more balanced interpretation of the data, particularly in view of the limited strength and breadth of the evidence. The reviewers also request additional information on the methodology used and the omission of pre-planned analyses, and provide guidance to strengthen the presentation and enable a more impactful consideration of the field and the implications of your conclusions. After discussing the paper with the editorial team and an academic editor with relevant expertise, I'm pleased to invite you to revise the paper in response to the reviewers' comments. We plan to send the revised paper to some or all of the original reviewers, and we cannot provide any guarantees at this stage regarding publication.

We ask that you submit your revision by Dec 12 2025 11:59PM. However, if this deadline is not feasible, please contact me by email, and we can discuss a suitable alternative.

Don't hesitate to contact me directly with any questions (afarrell@plos.org).

Best regards,

Alison

Alison Farrell, Ph.D.

Senior Editor

PLOS Medicine

afarrell@plos.org

Comments from the academic editor:

The authors present the use of RDTs in an overly positive light without adequately acknowledging the substantial limitations of the available data. In this context, I am not convinced that the studies included in the review can fully answer the question. Modelling studies may provide more informative insights, particularly regarding the potential impact on INSTI resistance. The risk of resistance is what motivated the review, yet this crucial outcome is not considered.

Regarding test accuracy, the authors limited their search to the LAI-PrEP context, but there is a substantial existing literature, including numerous studies and systematic reviews/meta-analyses, evaluating the accuracy of various RDTs compared with NAT or laboratory-based immunoassays. This evidence should be discussed in greater depth. As reviewer 2 notes, accuracy varies considerably across different RDTs.

Comments from the reviewers:

Reviewer #1: This is a well-written manuscript on an important topic -- what type of HIV testing might best be used in assessing HIV infection status in persons on long-acting injectable pre-exposure prophylaxis (PrEP). The authors have undertaken a systematic review of rapid, laboratory-based immunoassay and nucleic acid testing strategies. For the most part, the paper is balanced, the authors acknowledge the severe limitations of the current data available. However, in some spots, it seems they under-emphasize some of the advantages of laboratory and/or NAT testing. They also speak to the advantages of self-testing, which is not at all addressed by the current set of studies. While I agree with their premise that overall the benefits of RDT outweigh the downsides, the authors could be more balanced in their treatment of the data.

Major comments:

1. The authors state the benefits of the speed of results with RDTs compared to NAT and lab-based assays. However, this is generally a matter of several days difference, whereas the more sensitive NAT testing may pick up infections months earlier. In terms of speed of getting information on HIV status for the rare breakthrough infections, it seems that NAT would be the winner.

2. The authors cite the proportion of delayed infections using the denominator of all persons in the study (line 278) rather than just of those found to be HIV positive. Should the denominator be the latter, rather than the former?

3. In citing differences in resistance-associated mutations, the authors state, "...there is no direct evidence that the testing strategy would affect the development of resistance-associated mutations" (lines 311-313). However, the data do suggest that 80% of the infections in HPTN 083 suggested that NAT would have picked up the resistance prior to emergence of INSTI RAMs.

4. The authors talk about the benefits of oral self-testing, but none of the data evaluated this method of diagnosis, and the window period for oral testing is considerably longer (~12 weeks) than even RDT and certainly than lab-based and NAT testing. The reference to oral testing seems to be misplaced in this description of the literature.

5. Is the turnaround time for NAT testing really 35 days (line 367)? That seems quite long compared to what is available in clinical practice.

6. The authors also state that RDT allows for same-day starts, a very important component. However, most guidelines do allow for screening with RDTs and then confirmation with lab-based testing, which wouldn't actually rule out same-day starts. The authors might comment on whether or not confirmatory testing is needed, rather than stating that testing other than RDTs would necessarily rule out same-day starts.

Minor comments:

1. Line 108: the authors refer to IAS guidelines -- did they intend this to be IAS-USA guidelines?

Reviewer #2: Thank you for your recent submission assessing long acting injectable PrEP for HIV patients. From a methodological standpoint the review is conducted using gold standard methods. A highlight being the extensive literature searching.

The analysis of odds ratio data was conducted using the PM fixed effect model, which seems appropriate due to small/no events and lack of heterogeneity. However, your protocol states you planned to conduct a bivariate analysis of the sensitivity and specificity for diagnostic accuracy. This does not appear to have been conducted and no reasoning appears to be presented. It seems potentially feasible for LA-CAB with five studies being available. If the authors deemed it unfeasible, could they please add detail to state why it was not.

Could the authors provide further information on how GRADE was conducted. Especially, where evidence was meta-analysed. Did the authors use the guidelines by Murad and colleagues (2017: https://doi.org/10.1136/ebmed-2017-110668)? If not, the authors may wish to consider this methodology where they do not have a single point estimate. Additionally, if they did use this method this should be stated clearly with the appropriate reference.

I think the discussion, at times, is a little over positive with the findings of the review. It should emphasis that there is a lack of evidence and the quality of the evidence is low/very low. This is mentioned for some outcomes but I feel this may stretch across all outcomes given the current GRADE assessments, risk of bias, and sparsity of data.

Reviewer #3: This systematic review aims to assess the effects of different testing strategies to support long-acting injectable pre-exposure prophylaxis roll-out. The strength of the paper is the considerable manuscripts and abstracts that were reviewed. However, I major limitation is the conclusion that rapid diagnostic tests should be used despite low certainty of evidence, the evidence clearly favoring nucleic acid tests, a discussion of acute HIV infection and the detrimental effects of starting someone on LAI-PrEP when they really need treatment and the further potential transmission risks.

My main suggestions are that as follows:

1) Replace table 1 which discusses the main characteristics of the study with a table that lists every paper that was included in the review (n=22) with details about a) authors b) publication date c) country of study d) test used e) study design f) type of PrEP g) study population h) sample size i) outcome of interest k) main result. Include a similar section for abstracts that were included.

2) In table 2, it would beneficial to the reader to see the studies from which these findings came and how each of those studies were classified using GRADE

3) Some discussion of the different generations of enzyme immunoassays for HIV and the window of detection is warranted as well as the importance of acute HIV infection and challenges of diagnosis with earlier generation tests. A fourth-generation assay would detect acute HIV but the earlier (1st, 2nd, 3rd) will likely miss infections. This is an important consideration as resistance can be bred if an infection person if put on monotherapy. Here a discussion of repeat testing if using older generation and a risk assessment to understand recent exposures is important.

4) Please describe in more detail the approach to the systematic review using PRISMA and include supplemental tables as well as the checklist. For example, please include a table of inclusion and exclusion criteria. See Liberati A, Altman DG, Tetzlaff J, et al. The PRISMA statement for reporting systematic reviews and meta-analyses of studies that evaluate health care interventions: explanation and elaboration. PLoS Med2009;6:e1000100. doi:10.1371/journal.pmed.1000100 pmid:19621070

5) The forest plot clearly shows that the data favor NAT. In a high-risk population, it is even more important to ensure a client's HIV negative status before starting PrEP. For the general population and HIV screening programs, the diagnostic accuracy might be compatible but is that true for high risk populations? Aren't their special considerations for testing persons at high risk of HIV acquisition? See: https://www.sciencedirect.com/science/article/pii/S1386653212000467?via%3Dihub

6) How will being on PrEP affect our ability to make an HIV diagnosis and does that also warrant consideration in choosing which test to use in PrEP programs for initiation and follow-up? See: https://pmc.ncbi.nlm.nih.gov/articles/PMC6918508/. An algorithm might be an appropriate approach for PrEP program especially if using RDTs. Recommendations for repeat testing might be needed and evaluation for resistance if a patient is infected and missed by RDT and started on PrEP OR they seroconvert on PrEP. The latter would be extremely rare with LAI-PrEP.

7) The cost discussion is interesting but putting it into the context of using an RDT and missing a diagnosis - the cost of PrEP, delayed diagnosis, potential resistance, potential further transmission - should be considered too.

---

* Please upload any figures associated with your paper as individual TIF or EPS files with 300dpi resolution at resubmission; please read our figure guidelines for more information on our requirements: http://journals.plos.org/plosmedicine/s/figures. While revising your submission, we strongly recommend that you use PLOS's NAAS tool (https://ngplosjournals.pagemajik.ai/artanalysis) to test your figure files. NAAS can convert your figure files to the TIFF file type and meet basic requirements (such as print size, resolution), or provide you with a report on issues that do not meet our requirements and that NAAS cannot fix.

After uploading your figures to PLOS's NAAS tool - https://ngplosjournals.pagemajik.ai/artanalysis, NAAS will process the files provided and display the results in the "Uploaded Files" section of the page as the processing is complete.

If the uploaded figures meet our requirements (or NAAS is able to fix the files to meet our requirements), the figure will be marked as "fixed" above. If NAAS is unable to fix the files, a red "failed" label will appear above.

When NAAS has confirmed that the figure files meet our requirements, please download the file via the download option, and include these NAAS processed figure files when submitting your revised manuscript.

* Please confirm that all authors have declared any COIs.

* Please note that we require the inclusion of funders' websites (URLs).

* Please ensure that the study is reported according to the appropriate guideline and include the completed checklist as Supporting Information. When completing the checklist, please use section and paragraph numbers, rather than page numbers. Please add the following statement, or similar, to the Methods: "This study is reported as per PRISMA guideline (S1 Checklist)."

FIGURES AND TABLES

SUPPLEMENTARY MATERIAL

REFERENCES

SYSTEMATIC REVIEWS & META-ANALYSES

* Please report your SR/MA according to the PRISMA guidelines provided at the EQUATOR site. http://www.equator-network.org/reporting-guidelines/prisma/. Please provide the completed PRISMA checklist as Supporting Information. When completing the checklist, please use section and paragraph numbers, rather than page numbers. Please add the following statement, or similar, to the Methods: "This study is reported as per the Preferred Reporting Items for Systematic Reviews and Meta-Analyses (PRISMA) guideline (S1 Checklist)."

* Abstract: Please report your abstract according to PRISMA for abstracts (https://doi.org/10.1371/journal.pmed.1001419) following the PLOS Medicine abstract structure (Background, Methods and Findings, Conclusions). Please ensure you provide dates of search, data sources, number of studies included, types of study designs included, eligibility criteria, and synthesis/appraisal methods.

* Please note that we expect searches to be updated to within 6 months of the time of submission.

---

## [Decision Letter · Decision Letter 2]

5 Mar 2026

Dear Dr. Ong,

Thank you very much for re-submitting your manuscript "Balancing Accuracy, Accessibility, and Cost: HIV Testing Strategies for Long-Acting Injectable Pre-Exposure Prophylaxis: A Systematic Review of Rapid, Laboratory-Based Immunoassay and Nucleic Acid Testing Strategies" (PMEDICINE-D-25-02056R2) for review by PLOS Medicine.

I have discussed the paper with my colleagues and the academic editor and it was also seen again by 2 reviewers. I am pleased to say that provided the remaining editorial and production issues are dealt with we are planning to accept the paper for publication in the journal.

[LINK]

We look forward to receiving the revised manuscript by Mar 12 2026 11:59PM.

Sincerely,

Alison Farrell, Ph.D.

Senior Editor

PLOS Medicine

plosmedicine.org

Requests from Editors:

* Please review your text for claims of novelty or primacy (e.g. 'for the first time') and remove this language. In addition, please check that any use of statistical terms (such as trend or significant) are supported by the data, and if not please remove them.

* In the author summary, in the final bullet point of 'What Do These Findings Mean?', please include the main limitations of the study in non-technical language.

* Please provide titles and legends for all figures and tables (including those in Supporting Information files). Please define all acronyms used in each figure or table in its corresponding legend.

* Please ensure that where relevant figures include 95% CIs.

* Please ensure that all abbreviations are defined at first use throughout the text.

* Please confirm that all numbers presented in the abstract are present and identical to numbers presented in the main manuscript text.

* Please confirm that your title complies with PLOS Medicine's style. Your title must be nondeclarative and not a question. It should begin with main concept if possible. "Effect of" should be used only if causality can be inferred, i.e., for an RCT. Please place the study design ("A randomized controlled trial," "A retrospective study," "A modelling study," etc.) in the subtitle (ie, after a colon). We suggest: “Laboratory-Based Immunoassay and Nucleic Acid HIV Testing Strategies for Long-Acting Injectable Pre-Exposure Prophylaxis: A Systematic Review

* It appears that one or more study authors is affiliated with one or more of the agencies that funded the study (WHO). Thus, the statement “The funders had no role in study design, data collection and analysis, decision to publish, or preparation of the manuscript” does not apply. Please revise the Financial Disclosure accordingly, as in "[Author name] is [author's role] at [funding agency]. The funders had no other role in study design…..”

Please provide the funder URL(s)

Please indicate what LAI-PrEP is for in the first sentence of the Background (i.e. prevention of HIV acquisition)

Please clarify in the Abstract and Methods that you included studies to April 2025—based on the rebuttal information

Line 52: please remove period after methodology.

Please explain in the Abstract what the 8171 and 2483 denominators are.

Line 70: change ‘ensuring’ to ‘which can ensure’

Line 73: delete ‘,with self-testing as an option’, as per reviewer comment.

In the Abstract, a clearer explanation of why testing might differ for LAI vs oral PrEP is needed in the Background.

Please clarify definition for HIVST ( used to define both the test and self-testing but defined as self-testing)

Line 82: what is meant by ‘early HIV infections’?

Author summary: please clarify the distinction between a lab-based test and a NAT. Suggest to use laboratory-based immunoassays (if correct) throughout to avoid confusion.

This statement, line 154, “This review extends beyond comparing the analytical accuracy of HIV tests” is not necessary in either the Abstract or the Introduction.

Persons not funders should be included in the Acknowledgments.

Please provide a clean version of the Supplementary material

Please be advised that in general we require that systematic review searches are updated to within 6 months of final submission. In this case we feel it would be worth performing the search and adding a sentence indicating the results (i.e. have there been additional relevant studies since April 2025 or not?) for full transparency for readers.

Please make sure the manuscript metadata matches the manuscript.

"* Please report your SR/MA according to the PRISMA guidelines provided at the EQUATOR site.

http://www.equator-network.org/reporting-guidelines/prisma/

When completing the PRSIMA checklist, please use section and paragraph numbers, rather than page numbers (page numbers are presently used).

Please add the following statement, or similar, to the Methods: " "This study is reported as per the Preferred Reporting Items for Systematic Reviews and Meta-Analyses (PRISMA) guideline (S1 Checklist)." " "

* Please provide the beginning and end dates of your search.

* Please report your abstract according to PRISMA for abstracts, following the PLOS Medicine abstract structure (Background, Methods and Findings, Conclusions) http://www.plosmedicine.org/article/info:doi/10.1371/journal.pmed.1001419 .

Comments from Reviewers:

Reviewer #1: The authors have responded thoroughly and persuasively to the majority of comments by the reviewers. However, a few issues remain:

1. The authors still insert in the conclusions of the abstract that self-testing should be considered, when their review (and their data) do not support this recommendation.

2. Lines 141-143: The issue isn't so much about breakthrough infections when the levels of the LAI are low, but rather giving people monotherapy who have undetected infection at inception of LAI.

3. Throughout, there is mostly discussion of NAT vs. RDT. What about confirmatory lab-based Ag/Ab testing that is not mandatory? What is the cost of that testing, and hwo would that affect the cost calculations (lines 389-390 and Table 2)?

Reviewer #3: Thank you for addressing my comments so thoroughly. I do still think that the tables in the Supplemental information - namely S4 and S5 should be included in the main paper. However, I defer to the editors as they might have a preference given the size of the tables. These data are important to share so perhaps links to the supplement could be included when they are referenced in the main paper.

[LINK]

---

## [Editor Report · Decision Letter 3]

18 Mar 2026

Dear Dr Ong,

On behalf of my colleagues and the Academic Editor, Matthias Egger, I am pleased to inform you that we have agreed to publish your manuscript "Rapid Diagnostic Tests, Laboratory-Based Immunoassay and Nucleic Acid Testing Strategies for Long-Acting Injectable Pre-Exposure Prophylaxis: A Systematic Review" (PMEDICINE-D-25-02056R3) in PLOS Medicine.

Please also note that we require a declaration of competing interest statement from all authors.

PRESS

Sincerely,

Alison Farrell, Ph.D.

Senior Editor

PLOS Medicine